

# Facing policy challenges with inter- and transdisciplinary soil research focused on the SDG's.

Johan Bouma[1] and Luca  Montanarella[2].

[1]Em.prof soil science, Wageningen University, the Netherlands

[2]Senior Expert, Joint Research Centre, European Commission, Italy

## Abstract

Our current information society, populated by increasingly well informed and critical stakeholders, presents a challenge to both the policy and science arena's. The introduction of the UN Sustainable Development Goals offers a unique and welcome opportunity to direct joint activities towards these goals. Soil science, even though it is not mentioned as such,  plays an important role in realizing a number of SDG's focusing on food, water, climate, health, biodiversity and sustainable land use. A plea is made for a systems approach to land use studies, to be initated by soil scientists, in which these land-related SDG's are considered in an integrated manner. To connect with policy makers and stakeholders two approaches are functional, following: (i) the policy cycle when planning and executing research, which includes *signaling, design, decision, implementation and evaluation.*Many current research projects spend little time on *signaling* which may lead to disengagement of stakeholders. Also, *implementation* is often seen as the responsibility of others while it is crucial to demonstrate – if successful- the relevance of soil science and (ii) the DPSIR approach when following the policy cycle in land-related research, distinguishing external *drivers, pressures, impacts and responses* to land-use change that affect the *state* of the land in past, present and future.Soil science cannot by itself realize SDG's and interdisciplinary studies on Ecosystem Services (ES) provide an appropriate channel to define contributions of soil science in terms of the seven soil functions. ES, in turn, can contribute to addressing the six SDG's ( 2,3,6,12, 13 and 15) with an environmental, land-related character. SDG's have a societal focus and future soil science research can only be successful if stakeholders are part of the research effort in transdisciplinary projects, based on the principle of time-consuming "joint-learning".The internal organization of the soil science discipline is not yet well-tuned to the needs of inter- and transdisciplinary approaches.





**List of abbreviations**
CAP      Common Agricultural Policy
CBD      Convention on Biological Diversity
DPSIR    Drivers, Pressures, State, Impact, Response related to land use change
EC       European Commission
ES       Ecosystem Services
EU       European Union
GSP      Global Soil Partnership
IPBES    Intergovernmental Platform for Biodiversity and Ecosystem Services
IPCC     Intergovernmental Panel on Climate Change
ITPS     Intergovernmental Technical Panel on Soils
MEA     Multilateral Environmental Agreements.
SDG     Sustainable Development Goal
UNFCCC UN Framework Convention on Climate Change
UNFCCC UN Framework Convention on Climate Change






**Introduction**
This paper will discuss the relationships between policy and sustainability research
focusing on soil science,realizing that societies have been subject to major changes
in the recent past.Fifteen years ago, the internet had hardly established itself.Now,
billions of people have computers and mobile phones and unlimited access to an
overwhelming quantity of information via the World Wide Web.Scientists are not the



only source of information anymore as they were in the not too distant past,at least in
their own perception.Rather than deliver information by communicating results of
their research they are now increasingly faced with the challenge to judge information
provided by the Web and channel it to interested stakeholders.Also, stakeholders
become more knowledgeable and critical.A recent analysis showed that more than
50% of young Dutch farmers has a BSc or MSc degree.After all, many of them are
our own students!
These societal changes not only had a major impact on the policy arena,where
citizens become more active outside the traditional political party systems,but also on
the relation between science and society.Rather than be just recipients of
information,citizens are increasingly partners in joint learning processes.This not only
applies to so-called developed countries but increasingly to developing countries as
well where mobile phones are the primary source of an information revolution.It
appears that the soil science community ,like other disciplines,is struggling to catch
up with these modern developments as many traditional procedures in this
profession,established in the 19th century,appear to be rather strongly entrenched.
The effects of societal changes on policy and science will be discussed with the
objective to explore future possibilities for creative and productive interactions
between the policy and scientific arenas,with particular attention for the role of soil
science research when presenting effective contributions towards the achievement of
sustainable development goals.
**The policy arena: science meeting society.**
A policy is a statement of intent and a deliberate system of principles to guide
decisions and achieve rational outcomes after implementation.The policy cycle
consists of a number of phases (e.g. Althaus et al, 2007, Bouma et al, 2007): (i) the
*signaling* phase in which problems are identified, based on a characterization of
current conditions; (ii) the *design* phase in which options for possible corrective action
are defined based on research using existing and newly acquired information; (iii) the
*decision* phase in which a selection is made by policy makers of options being
presented.Here,negotiation processes play an important role; (iv) the *implementation*
phase in which the selected option is being realized,and (v) the *evaluation* phase in
which the entire process is  analysed in terms of a learning procedure,applied to all
participants.This may have to include monitoring procedures to document





achievements.To be effective, all phases of the policy cycle require some form of
interaction between stakeholders involved, governmental agencies,policy makers and
scientists.A good example is certainly the US Soil Conservation Act of 1935,
responding to the severe soil degradation processes leading to the well-known "Dust
Bowl" syndrome that caused serious economic and social problems in that historical
period of the United States.But soil related policies have only rarely completed the full
policy cycle as described above.In Europe the attempt to reach the implementation
phase of the proposed EU Soil Framework Directive was ultimately stopped by the
lack of political will of some EU Member States to go beyond the negotiation and
decision phase.
Policies can be pro-active and reactive, but the latter usually applies. An example is
the Nitrate Directive (ND) (EC,1991) that was initiated because of very high nitrate
concentrations in groundwater in many European countries,following excessive
fertilization practices in agriculture.A water quality threshold of 50 mg nitrates/litre
had already been established in literature.It would have been most logical to require
measurements of nitrate concentrations in groundwater at different locations, to
compare these values with the threshold and next conclude whether or not quality
was adequate.However,measurements of nitrate concentrations in water were
cumbersome at the time, costly and time consuming and data were hardly available.
As any policy measure needs to be organized in such a way that operational
procedures can ensue,an alternative "proxy" was selected in terms of a maximum
fertilization rate of organic manure corresponding with 170 kg N/ha (e.g. Bouma,
2011). This corresponds with the manure production of appr. 1.7 animals/ha which
can be easily controlled by regulators because the number of animals and ha's are
known for each farm. Groundwater quality in the late 1980's was considered to be
quite poor in many areas and measures had therefore to be taken quickly: the
*signaling, design, decision* and *implementation* phases of the policy cycle followed
very rapidly.The 170 kg N/ha was not based on research, relating different
application rates of fertilizers to nitrate enrichment of groundwater as a function of
weather and soil conditions but was essentially empirical in nature.Science played a
role only as problem recognizer,documenting high nitrate contents of groundwater.
After 25 years,this policy has been quite successful in the Netherlands.Average
nitrate contents in groundwater in sandy soils were 190 mg/l in 1991 which was way



above the critical threshold.After introduction of the ND in 1991,contents have
gradually decreased and in 2012 the average content corresponded with the
threshold.However, contents in sandy soils were lower than the threshold in the
Northern part of the country and are still higher in the southern part.Nitrate contents
in clay soils were still 80 mg/l in 1998 but decreased to 20 mg/l in 2012,while
contents in peat soils were always lower than the threshold.Loess soils in the
southern tip of the country had higher contents than 50 mg/l in 2012 but these soils
only occupy a small area and their very deep watertables create quite different
conditions *(www.rivm.landelijk_meetnet_effecten _mestbeleid* ).Other problem areas,
such as the quality of surface waters and  nature areas, are discussed elsewhere (
Bouma, 2016 ).Possibly due to  the apparent success of the ND,there has not yet
been attention for an in-depth *evaluation* phase of the policy cycle and this will be
discussed later in more detail.
Restricting attention to the ND,should the role of science be different in future, and, if
so, why?
**The changing roles of science and policy in the information society.**
The internet was only present in rudimentary form in 1991.Now,everybody is
connected to the internet by computer or mobile phone and this is also true for many
developing countries.The world-wide-web creates an enormous flow of information
and scientists are increasingly engaged in interpreting and screening information that
reaches and often confuses users, stakeholders and policy makers alike.At the same
time well educated users ask ever more pertinent and critical questions.The roles of
the various participants in the societal debate that seemed rather well defined even
thirty years ago,have fundamentally changed.Authority is gained by the quality of
what is presented,not by the position of the presenters.Some see contributions of
science as: "just another opinion" and feel that science has to regain its: 'license to
operate".How to deal with this?And how do these effects influence policy makers?
Confronted with citizens of the Knowledge Democracy ( In't Veld, 2011) and battered
by social media that react instantly to policy measures,and preferably to policy
failures,policy makers and regulators become highly risk averse,avoiding controversy
if at all possible.This does not invite introduction of innovative measures nor definition
of clear goals for future action which may be controversial.Also, there is a tendency in
many western countries to decentralize decision making providing more



responsibilities to regional, provincial or communal entities.Scientists not only face
therefore more knowledgeable and critical stakeholders but also a more diverse
group of policy makers.How to deal with this and how to turn these new conditions
into an advantage by disruptive thinking, focusing on innovation? (e.g. Loorbach and
Rotmans, 2010;Schot and Geels, 2008)..A successful example of close linking of the
scientific advice and the policy making process is certainly the climate change policy
arena.Here the main driver has been the well recognized role of the
Intergovernmental Panel on Climate Change (IPCC) in providing high level policy
relevant scientific advice through highly reliable assessments.This role of IPCC has
gained the members the well deserved Nobel Prize in 2007.The strength of IPCC is
that, while being an intergovernmental body nominated by governments, it retains a
very high scientific credibility also within the scientific community.This allows IPCC to
deliver assessments that are fully endorsed by the related scientific community and
fully accepted by the policy making community as well.Such a crucial role of acting as
a science-policy interface has been identified as urgently needed also for other
multilateral environmental agreements (MEA's), like CBD (Convention on Biological
Diversity) and UNCCD (Convention to Combat Desertification in Africa).The recently
established Intergovernmental Platform for Biodiversity and Ecosystem Services
(IPBES) has indeed the ambition to serve like IPCC as the science policy interface
for CBD and also for other related MEAs.The need for such a science-policy interface
also for soils was well recognized in 2011 during the negotiations for the
establishment of the Global Soil Partnership (GSP).Indeed within the GSP the
Intergovernmental Technical Panel on Soils (ITPS) has been established and is
already operating since three years.It's first assessment will be the Status of World's
Soil Resources report, released at the closing ceremony of the UN International Year
of Soils 2015.

**185  Signaling as a crucial element of the policy cycle focusing on the SDG's.**

Despite all societal changes that soil scientists are confronted with,the policy cycle
still applies.*Signaling* requires definition of goals and an assessment as to whether
current conditions allow goals to be reached when proper measures are taken or
when this will not be possible defining drastic change.The recent 17 UN Sustainable
Development Goals (Table 1) (http://sustainabledevelopment.un.org/focussdgs.html) provide a
valuable point of reference for the policy cycle and for *signaling* in particular.Soils are



not an SDG goal by themselves but they have a strong relation with health ( SDG 3),
water (SDG 6),climate (SDG 13) ,biodiversity (SDG 15) and sustainable development
(Several SDG's, for soil science particularly SDG 15 which mentions land
degradation).All these goals cannot be reached by just studying soils but require
interdisciplinary approaches, including contributions by soil science that often have a
significant effect on results.For example,Bonfante and Bouma (2015) used soil maps
and simulation modeling to assess the spatial effects of irrigation practices on the
growth of eleven maize hybrids,considering effects of climate change.Results allowed
more efficient targeting of water allocation and choice of hybrids for different soil
conditions.This was new and surprising for the hydraulic engineers and plant
breeders involved who had a rather traditional and static image of the soil science
profession.The example shows the advantage of reaching out to other professions.
More examples are available and they should be communicated more clearly,
demonstrating interdisciplinarity in practice.
SDGs are globally applicable and will have to be implemented during the next years
by all National governments.Of crucial importance will be the way in which progress
towards achieving each goal will be measured.The adoption of an agreed set of
indicators becomes therefore of fundamental relevance for the implementation and
evaluation phase of the SDGs.Introducing soil related indicators for the SDGs that
explicitly mention soil as a component  would be desirable, but will face the well
known lack of basic soil data and adequate soil monitoring systems in many Nations
of the world.A more realistic approach will be to use proxy indicators adressing the
goals in a more holistic and integrated manner.
In general, the ecosystem services (ES) concept is suitable to express this
interdisciplinary effort because disciplines by themselves cannot define ES. (Table 2)
(De Groot et al, 2002, Dominati et al, 2014).The next step is to define the role of soils
in contributing to the provision of ES and then the seven soil functions of the EC (
EC, 2006) can be considered (Table 3).For example, SDG 2:*"End hunger, improve
*nutrition and promote sustainable agriculture"*  relates to the provisioning ES 1,
relating to food. But sustainable development also requires regulating ES 5, 6,7 and
8.Soil functions 2,3 and 6 define the contributions that soil science can make to these
more general ecosystem services, which, again, not only require an inter- but also a
transdisciplinary approach.Bouma et al ( 2015) presented six transdisciplinary case



studies, identifying relevant SDG's,ES and soil functions as an example of framing
based on studies that were made and published in the past with a traditional scientific
focus.They also concluded that in three of the studies existing knowledge was
adequate to solve the problem being studied.In the remaining studies new research
was needed and defined based on observed gaps in existing knowledge. To avoid
confusion,it is important to refer to general ecosystem services and to soil
contributions towards those services to be articulated by the soil functions.Terms like
soil services or soil ecosystem services should be avoided.
**The DPSIR system**
When studying SDG's, ES and  the application of soil functions in the context of the
policy cycle, the DPSIR system, (Van Camp et al, 2004, Bouma et al, 2008) is helpful
to analyse processes involved ( Figure 1). Here, S represents the state of the land;D
represents drivers of land use change,P are the resulting pressures on the land,I is
the impact,and R,finally,indicates a respons in terms of development of strategies
and operational procedures for the mitigation of perceived threats.The flowchart in
Figure 1 shows the past, present, and future state S of the land.Drivers and
pressures in the past have led to impacts and, most likely, certain responses.This all
results in a present state S which is not only determined by soil factors but can be
defined by the ecosystem services it can provide by mobilizing relevant soil functions.
This dynamic characterization of the state S is preferred over a static one applying,
for instance, a set of soil characteristics as has been the traditional approach in land
evaluation (e.g. Bouma et al, 2012).
Of particular interest,of course,are future developments that are considered in terms
of different scenarios,each one associated with characteristic drivers,pressures and
impacts.Different scenarios  represent different visions on sustainability and have,of
course,only an   exploratory character.In the past scientists of different disciplines
acted rather independantly when assessing the various components of the DPSIR
system and when defining scenarios,but today soil scientists would be well advised to
interact and engage colleagues in other sciences,stakeholders and policy makers
during the evaluation period to make sure that all options are considered and that
their input is taken into account.This requires a truly transdisciplinary process (e.g.
Thomson-Klein et al, 2001).The combined scenarios,presenting a series of
alternative options, are presented to the policy arena.Selection has to be made by



politicians and citizens, **not by scientists**.This is a crucial point because scientists
should maintain their independance and should not be seen as partners in the policy
arena or of certain business interests.Often risk averse politicians are more than
willing to escape their responsibilities and hide behind scientists,which can be
damaging to the scientific reputation.The described scenario approach,defining a
series of states S with all its attributes is therefore more appropriate than presenting
only one,"ideal" option as defined,for example,by a group of scientists.When
considering  sustainable  development,environmental,social,and  economic
considerations and approaches have to be mutually balanced to achieve some type
of compromise that is acceptable to a wide range of stakeholders (be it grudgingly
because their demands can only be partly met in the ultimate compromise) .Usually,
economic  considerations  largely  determine  the  outcome  of  this  type  of
interdisciplinary analysis.The scheme in Figure 1 suggests an approach where
environmental  and  social  aspects,expressed  by  DPIR,are  considered  first  and
economic considerations come later in terms of a cost–benefit analysis for each of
the Sf scenarios.The recently proposed Soil Security concept ( Mc Bratney and Field,
2015),distinguishing capability,condition,capital,connectivity and codification,fits into
the DPSIR scheme.The actual condition corresponds with S and also represents
capital.Capability is represented by the scenario's in figure 1,connectivity with the
required inter- and transdisciplinay approach and codification is the domain of
legislators being fed with relevant information.
This analysis indicates that the *signaling* phase of the policy cycle is very important
because the option being chosen in the end is,ideally,the result of an extensive
participatory process.If so,*design* can receive well focused attention and *decision and*
*implementation* can follow rather quickly and harmoneously.

**Science versus policy in the real world**

As discussed,the introduction of the ND after 1991 did not follow the ideal policy
cycle.*Signaling, design, decision and implementation* followed quickly because the
groundwater quality issue was considered to be critical. In retrospect,the soil science
community was succesful in the preceeding years  documenting the effect of different
fertilizer practices on groundwater quality but they paid no attention to what an
enforcable policy to overcome the problem might look like.Policymakers had to act on



their own.After 24 years the policy is unchanged, while many questions are being
raised.The universal application rate of 170 kg N/ha does no justice to different
processes in different soils and to effects of management. Examples are found where
much higher application rates result in low nitrate contents in groundwater.In fact, the
ND becomes a defacto means to restrict intensification of agriculture, which is a
much broader policy goal (with major societal implications) than groundwater quality.
Stakeholders are aware of this and even though well educated farmers support
measures to enhance environmental quality, they resist "policy drift", when objectives
secretly change in time.Also,they question what appear to be seperate regulations for
groundwater,surface water,air and nature quality while nutrient regimes are obviously
related to all of them: nitrogen that moves into groundwater cannot be emitted to the
air.(e.g. Bouma, 2016).Recent studies for Dutch dairy farms took a systems approach
by applying a Life Cycle Assessment for the entire farming operation,not only
covering the emission of nutrients to both air and water but net income and energy
use as well (Dolman et al, 2014;De Vries et al, 2015).A group of eight farmers
followed a nutrient cycling approach to reduce fertilizer use and results of their
farming operations were compared with a control group.The program was highly
interactive, involving intensive contact with farmers, demonstrating a good example
of inter- and transdiciplinary researchThere was time for *signaling, design and*
*decisions* by cooperating scientists and farmers,followed by *implementation.*The
entire procedure took about 20 years.Farmers,following the nutrient cycling
approach,had lower use of fertilizer and energy ,lower emissions and higher net
incomes and organic matter contents of their soils due to management.But due to the
high variability among farms,only energy use and organic matter contents were
significantly different when compared with a control group of eight farms.Rather than
focus on average values for a group of farmers it would in retrospect have been
preferable to focus on individual farms because every farm "has a different story to
tell".
Droogers and Bouma (2012) studied accelerating future water shortages in Asia and
Africa , requiring development of operational water governance models, as illustrated
by three case studies: (1) upstream–downstream interactions in the Aral Sea basin,
where the *signaling* function of science was most prominent;  (2) impact and
adaptation of climate change on water and food supply in the Middle East and North
Africa, where not only *signaling* was important but also a broad *design* and a timid



start *of implementation* and (3) Green Water Credits in Kenya, where the entire policy
cycle was covered, including the start of *implementation.* (Kauffman et al, 2012).

**328 From signaling to implementation**

Any impression that the sequence of *signaling* all the way to *implementation*
represents a smooth ,sequential process is,unfortunately, misleadingly simple. A
major study on sustainable agriculture in the Netherlands showed that interactions
between researchers, various stakeholders and policy makers were complex and
repetitive, which can be shown in a diagram visualizing interaction processes. Figure
2 (from Bouma et al, 2011) illustrates this for case study 1 in Dutch dairy farms, the
same study as the one mentioned above. *Implementation* could in the end only be
achieved because the farmers involved,assisted by soil scientists,persisted against
all odds.Kauffman et al ( 2012) presented comparable diagrams for the Kenya study.
The role of scientists in the *implementation* phase is different from the role in the
*signaling* and *design* phase.In the latter,all opinions are welcome,as described
above.But when plans and decisions have been made, *implementation* is a clear goal
and distractions are rather unhelpful.Soil scientists can play an important role here by
keeping the ultimate goal of the project in focus.It is also in their interest that specific
results are obtained to document the beneficial effect of their input.Designs on paper
of what appear to be most thoughtful and inventive projects have no impact and
create no credit for all involved when they are not realized.
There are in Europe already existing soil-related policy instruments that are
unfortunatly lacking the necessary scientific backup and support from the soil science
community.The most relevant example is the Common Agricultural Policy (CAP),
probably one of the most important (at least in monetary terms) policy of the
European Union.Obviously, there are major implications for soils when this policy is
fully implemented.The mandatory requirement for good agricultural and ecological
practices that farmers need to implement in order to access the direct payment
scheme of the CAP explicitly refers to soil parameters like soil erosion,organic carbon
and compaction.The correct implementation of such a cross-compliance scheme
should have a substantial impact on soil conditions across the EU.Unfortunately,
implementation has been rather weak and monitoring of the results by an



independent scientific community is essentially lacking.Soil scientists have missed an
opportunity to play a key role in this process.
Current projects leave little time for scientists to be seriously engaged with both
*signaling* and *implementation* and this may have to be changed in future considering
the demands but also the challenges and opportunities of the modern information
society (e.g. Bouma, 2015)*.*

**Soil science linking stakeholders and policy makers in the information society**

Changes in society,as discussed,have a strong impact on both the scientific and
policy arena.Both struggle to communicate well with modern stakeholders and to
define the role of science in the information age.When dealing with land-related
issues in the context of the SDG's,soil scientists are in an excellent position to
become effective intermediaries in the stakeholder-policy-science NEXUS for at least
two reasons: (i) traditionally soil scientists have worked intensively with stakeholders
in the context of soil survey or soil fertility studies,that involved extensive field work.
This has decreased as soil surveys were completed and fertility schemes became
well established. But traditions can be rejuvenated as a basis for truly
transdisciplinary research that can genuinely engage stakeholders and provide broad
support for policy measures,and (ii) even though soils are not mentioned in the
SDG's,they form a cross-cutting theme in issues that do receive attention:water,
climate, biodiversity (e.g.Montanarella and Lobos Alva,2015). This focus tends to
unintentionally enforce the disciplinary nature of the water, climate, and biodiversity
disciplines.Soil Science,related to " land" as no other discipline, can, in contrast, play
a pioneering role in initiating system studies that integrate the various issues in a
systems approach.Examples are the studies of Dolman et al, (2014) and De Vries et
al, (2015).This type of study  is attractive for stakeholders,like farmers,who have to
operate complex production systems and for policy makers focusing on
environmental quality,having to integrate seperate requirements of water,air and
nature.
One final aspect needs to be considered. The ND legislation in 1991 had a :"top-
down, command-and-control" character which was realistic at the time because
groundwater quality was poor in many locations and something had to be done





quickly.But after 25 years still the same top-down approach is followed at a time
when not only environmental conditions have significantly improved, but when also
the information society has drastically changed relations between policy and
stakeholders, as discussed.Bouma (2016) therefore argued for a new "bottom-up"
approach where tailor-made systems are designed for individual farms ,including
indicators that can be used for regulatory purposes.A "one-size-fits-all" approach
does not satisfy anymore at a time when well educated young farmers and other land
users have access to many tools and sensors that allow on-site characterization of
environmental conditions.

**Conclusions**

1.Traditional procedures in both science and policy are increasingly at odds with the
demands of the information society populated by well informed, critical stakeholders.
Soil scientists are in an excellent position to link the policy-stakeholder arenas when
dealing with land-related environmental issues,accepting the SDG's as common
goals.This will require not only inter- but also transdisciplinary research approaches
covering the entire policy cycle from *signaling* to *implementation*.
2.SDG's with an environmental focus can be approached by defining relevant
ecosystem services that require an interdisciplinary research approach including a
disciplinary assessment of the role of soil functions when contributing to these
ecosystem services.
3.Current research programs tend to emphasize the *design* phase of the policy chain.
More attention is needed for the *signaling* phase, where the DPSIR procedure can be
effective,as well as in the *design* phase. Attention for *implementation* is needed to
produce results supporting claims of relevance.
4."Top-down, command-and-control" environmental policy measures,as discussed
here for the Nitrate Directive should in time be replaced by:"bottum-up, interactive"
approaches fed by "tailor-made" designs for individual enterprises using inter- and
transdisciplinary research approaches.Only this approach is in line with the
requirements of the information society in the 21th century.

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
























**LIST OF TABLES**
Table 1 The seventeen UN Sustainable Development Goals
(http://sustainabledevelopment.un.org/focussdgs.html).
Goal 1 End poverty in all its forms everywhere
Goal 2 End hunger, achieve food security and improved nutrition and promote sustainable agriculture
Goal 3 Ensure healthy lives and promote well-being for all at all ages
Goal 4 Ensure inclusive and equitable quality education and promote lifelong learning opportunities for
all
Goal 5 Achieve gender equality and empower all women and girls



Goal 6 Ensure availability and sustainable management of water and sanitation for all
Goal 7 Ensure access to affordable, reliable, sustainable and modern energy for all
Goal 8 Promote sustained, inclusive and sustainable economic growth, full and productive
employment and decent work for all
Goal 9 Build resilient infrastructure, promote inclusive and sustainable industrialization and foster
innovation
Goal 10 Reduce inequality within and among countries
Goal 11 Make cities and human settlements inclusive, safe, resilient and sustainable
Goal 12 Ensure sustainable consumption and production patterns
Goal 13 Take urgent action to combat climate change and its impacts
Goal 14 Conserve and sustainably use the oceans, seas and marine resources for sustainable
development
Goal 15 Protect, restore and promote sustainable use of terrestrial ecosystems, sustainably manage
forests, combat desertification, and halt and reverse land degradation and halt biodiversity loss
Goal 16 Promote peaceful and inclusive societies for sustainable development, provide access to
justice for all and build effective, accountable and inclusive institutions at all levels
Goal 17 Strengthen the means of implementation and revitalize the global partnership for sustainable
development






Table 2 Ecosystem services (ES) with an important soil component according to
Dominati et al. (2014).
**Provisioning services**
1. Provision of food, wood and fibre.
2. Provision of raw materials.
3. Provision of support for human infrastructures and animals.
**Regulating services**



4. Flood mitigation
5. Filtering of nutrients and contaminants
6. Carbon storage and greenhouse gases regulation
7. Detoxification and the recycling of wastes
8. Regulation of pests and disease populations
**Cultural services**
9. Recreation
10. Aesthetics
11. Heritage values
12. Cultural identity












Table 3. The seven soil functions as defined by EC(2006)

1 Biomass production, including agriculture and forestry
2 Storing, filtering and transforming nutrients, substances and water
3 Biodiversity pool, such as habitats, species and genes
4 Physical and cultural environment for humans and human activities
5 Source of raw material



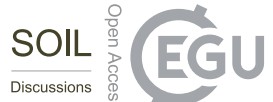

6 Acting as carbon pool
7 Archive of geological and archaeological heritage





















**List of figures**
Figure 1
Future land use scenario's (Sf)(derived in consultation with stakeholders, policy
makers and colleague scientists),  from which a choice has to be made in the policy



arena. Which one represents sustainable development best? (S=status of the land
defined in terms of the seven soil functions)



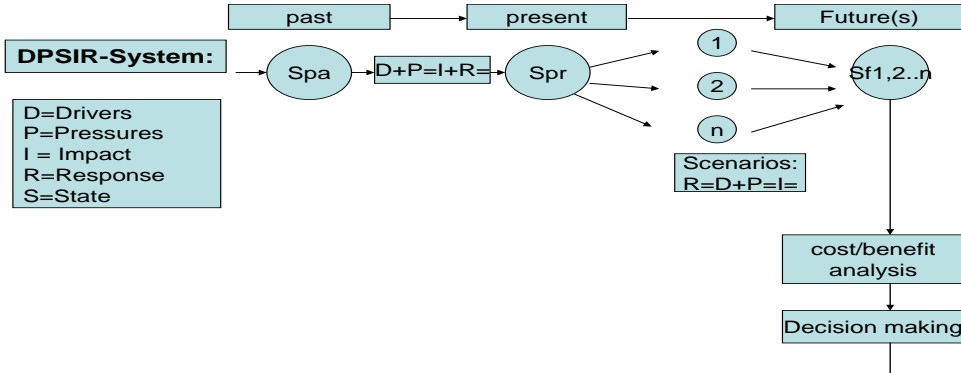













Figure 2
Schematic diagram showing complicated and long-duration interaction patterns
between different partners in a transdisciplinary study, developing a sustainable dairy
system in the Netherlands. N=NGO's; E= entrepreneurs; G= Government and K= the




knowledge arena. In this study ( Bouma et al, 2011), the policy cycle was simplified
here by describing *signaling* as *connected value proposition; design* as *-creation*
which includes *decision ,*while *implementation* corresponds with *- capture*.

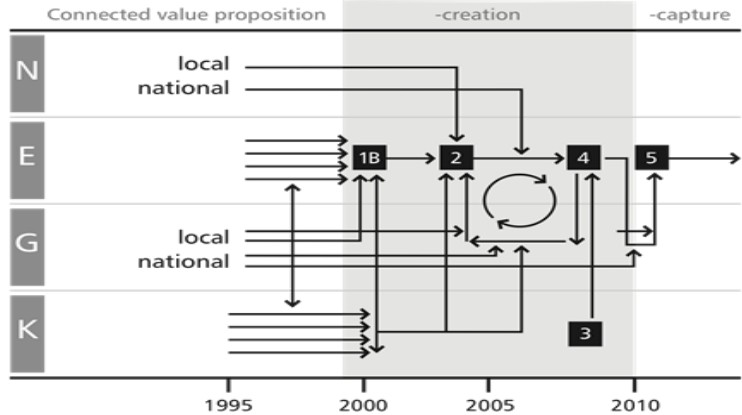
