# Peer review of "Published: 5 February 2016"

_SOIL, 2016_

## Referee Comment (RC1) · Anonymous Referee #1 · 16 Feb 2016

A very interesting manuscript analyzing a recent "hot" topic (SDGs). Find below some proposals for revising the manuscript:

L57: replace fifteen with twenty. It is almost 20 years that internet is a well-established mean of communication

In this first paragraph, please refer also to the contribution of "open" data or available "data" which have been increasingly been available during the last decade. Please refer to the work of Robinson (Science, 2015).

L66: "own students" : this limits only to the students of the authors. I would prefer a term which includes a broad audience.

L175: It is not only "Africa"

L182: remove "will" . The report is already been published. I would also propose to add a citation with a reference to the report.

L309: "." Is missing. 2 Sentences

L351: it is better to refer to "Good Agricultural and Environmental Conditions (GAEC)". Please replace the term Good agricultural and ecological practices with this one (official one in the CAP).

In this paragraph (L351-358), please take into account and cite the two recent published studies on the impact of Good Agricultural and Environmental Conditions (GAEC) of the CAP to reduces soil loss by water erosion in EU (Panagos, Borrelli, Robinson, NATURE, 596, 194) Moreover, it is also recommended to take into account the study of different agricultural practices to mitigate soil organic carbon (Lugato et al., 2014, Global change biology , 20 (11), 3557–3567).

General comment: It is very interesting the proposal to integrate soil scientists with ecosystem services. However, soil scientists should also be more integrated and work together with other disciplines such as hydrologists, earth science modellers, agronomists, natural hazards, economists (for food security issues), etc.

List of abbreviations (UNFCCC is listed twice, UNCCD is missing)

---

## Referee Comment (RC2) · Anonymous Referee #2 · 4 Mar 2016

This is a strong paper discussing and promoting inter- and transdisciplinary collaborations between soil scientists, other disciplines, and society in general.

Line 57 – By the mid to late 1990s I and pretty much everyone I knew were avid users of the internet. I would put the establishment of the internet at something more like 20 years ago, as opposed to 15. Hilbert and López (2011) support the ∼20 year time period.

Lines 63-64 "...stakeholders have become..."

Line 65 – The statement about the number of young Dutch farmers who have a BSc or MSc degree needs a reference.

Line 107 – "...established in the literature."

Line 139 – "...different in the future..."

Line 142 – Hilbert and López (2011) would also be a good reference for the idea the internet was in rudimentary form in 1991.

Line 153 – In't Veld should be 2010, not 2011.

Line 167 – "...role of IPCC gained..."

Lines 181-182 "...established and had been operating for three..."

Lines 186-205 – Adding one or more case studies to this manuscript that are not based on the work of the Bouma research group would help to strengthen the idea that this topic is important to people beyond the Bouma group. For example, another good case study would be that of Tabor et al. (2011), who used soil maps (among other tools) in a landscape-epidemiological study. Excellent example of transdisciplinary work that includes SDG 3.

There are 25 references in this manuscript, 12 of which were authored or co-authored by Bouma. Dr. Bouma has done excellent work, but the case that this is an important topic for soil science as a whole as well as allied fields and society would be strengthened if additional references (and case studies, as noted above) could be brought in to demonstrate a bit broader interest and activity within the soil science and related communities.

References Hilbert, M., López, P. The world's technological capacity to store, communicate, and compute information. Science 332, 60-65. 2011.

Tabor, J.A., O'Rourke, M.K., Lebowitz, M.D., Harris, R.B. Landscape-epidemiological study design to investigate an environmentally based disease. Journal of Exposure Science and Environmental Epidemiology 21, 197-211. 2011.

---

## Author Comment (AC1) · 19 Mar 2016

Reaction to reviewers comments on SOIL-2016-2 ( Bouma and Montanarella)

Reviewer1 We thank the reviewer for his positive reaction to this paper( "this is a very interesting manuscript analysing a recent "hot" topic"). In answer to his querries: 1. Line 57: Indeed, the internet is with us for 20 years! We will change this. 2. We will include reference to "open" data as suggested and to Robinson (2015). 3. We will refer to a broader audience than just our own students. 4. Line 175: correct; we will include a broader reference. 5. Line 182: we will include a reference to the published report. 6. L309: we will include the quotation marks. 7. Line 351: Indeed, since 2003 we should refer to Good Agricultural and Environmental Conditions (GAEC). This will be changed.

Good point! 8. Lines 3651-358. We will include the two suggested references as they are relevant in strengthening the discours. 9. We agree with the general comment on the role of soil scientists, but point out that being focused on ecosystem services, let alone on the Sustainable Development Goals of the United Nations, requires cooperation with other disciplines, the need for which is pointed out by the reviewer. We have incorporated this statement in the text of the revised manuscript and will refer to Keestra et al ( 2016) where this point is emphasized and illustrated.: Keesstra, S.D., J.Bouma, J.Wallinga, P.Tittonell, P.Smith, A.Cerda, L.Montanarella, J.Quinton, Y.Pachepsky, W.H.van der Putten, R.D.Bardgett, S.Moolenaar, G.Mol and L.O.Fresco. 2016. The significance of soils and soil science towards realization of the UN Sustainable Development Goals (SDG's).SOIL (doi:10.5194/soil- 2015.88). 10. Indeed, UNCCD ( United Nations Convention to Combat Desertification) is added.

Reviewer 2 We thank the reviewer for his statement that this is a "strong paper". In reaction to his comments ( excluding his grammatical suggestions that we will follow) : 1. Line 57: Yes, the internet is with us for a longer period than 15 years. We changed that . 2. Line 65: We will add a specific reference to a Dutch publication that documents the education levels of our young farmers. 3. Lines 186-205: yes, we will add some additional references. Aside from the suggested one by Tabor etal 2011, Italian work by Bonfante et al will be cited and two recent publications by Montanarella that include references to field work ( Nature, 528(7580), 32-3, 2015 and Current Opinion in Environmental Sustainability , 15, 41-48. 2015). Also, the Keesstra et al ( 2016) paper, cited above, contains relevant references.